# A Low-Density Polyethylene (LDPE)/Macca Carbon Advanced Composite Film with Functional Properties for Packaging Materials

**DOI:** 10.3390/polym14091794

**Published:** 2022-04-27

**Authors:** Jitladda Sakdapipanich, Phawasoot Rodgerd, Natdanai Sakdapipanich

**Affiliations:** 1Department of Chemistry and Center of Excellence for Innovation in Chemistry (PERCH-CIC), Faculty of Science, Mahidol University, Salaya Campus, Salaya, Phutthamonthon District, Nakhon Pathom 73170, Thailand; sayako29n@gmail.com; 2Mahidol Wittayanusorn School, Salaya, Phutthamonthon District, Nakhon Pathom 73170, Thailand; namounkung@gmail.com

**Keywords:** Macca carbon, composite films, anti-microbial activity, far-infrared, food packaging

## Abstract

Macca carbon (MC) powder, a biomass derived from macadamia nut cultivation that emits far-infrared (FIR) radiation, was incorporated into low-density polyethylene (LDPE) by melt-compounding and subsequent melt-extrusion operations. Optical microscopy, scanning electron microscopy, differential scanning calorimetry, thermal gravitational analysis, mechanical properties, FIR emission power, barrier properties, transmission properties, antimicrobial activity assays, and storage tests were used to evaluate the manufactured LDPE/MC composite viability sheets for antimicrobial packaging applications. The physical properties and antibacterial activity of composite films were significantly correlated with the amount of MC powder used. The higher the MC powder content in the LDPE/MC composite film, the better the FIR emission ability. Only the MC powder at 0.5% by weight displayed adequate fundamental film characteristics, antibacterial activity, and storage performance, allowing lettuce and strawberries to remain fresh for more than 7 and 5 days, respectively, outside the refrigerator. This study demonstrates that FIR composites made from MC powder are a distinct and potential packaging material for future application in the food industry.

## 1. Introduction

It becomes increasingly important to develop novel food packaging materials in recent years to protect food quality and safety and to extend the shelf life of food by reducing disease growth [1,2]. Polymer packaging that incorporates antimicrobial chemicals to prevent the growth of pathogenic and spoilage bacteria is an excellent packaging option [3,4]. Despite the fact that many products can act as antimicrobial agents, only a few commercialized products, such as silver-substituted zeolite, chlorine dioxide, ethanol, sulfur dioxide, and triclosan, are used in the food packaging industry due to strict safety and hygiene regulations, limited consumer acceptance of product effectiveness, and high costs [3]. Furthermore, consumers prefer goods that preserve food from natural sources in an effective manner. Several natural antibacterial compounds derived from plants and animals, including thyme, oregano oil, garlic oil, grapefruit seed extract, lactic acid, lysozyme, chitosan, and others, have been reported to be included in food packaging systems [5]. Although natural antimicrobial agents have low toxicity, their manufacturing costs are quite substantial [6]. Furthermore, many organic antibacterial compounds distort or evaporate when subjected to high processing pressure and temperature [2,7].

Consequently, inorganic synthetic materials have been established due to their capacity to survive severe processing conditions [3,8,9] and the unique functional features of food packaging. Through a photocatalytic process, TiO_2_ and ZnO may powerfully inactivate microbes [3,10]. The ceramic powder was utilized to develop commercial ceramic-filled polymer films with superior permeability [11]. The ceramic powder could emit far-infrared (FIR) radiation with wavelengths ranging from 6 to 14 µm, inducing biological systems to develop and flourish [12,13,14]. FIR radiation has the ability to permeate biological tissue and have a powerful rotational and vibrational energy effect at the molecular level without harming the skin’s surface [15,16]. The incorporation of ceramic powder into polyethylene and polypropylene films resulted in strong antibacterial activity and shelf-life extension of lettuces and strawberries [12,15]. It was claimed that the retained quality of strawberries wrapped in composite films was related to their FIR emission qualities, which impacted cell metabolism and aided exhausted cell recuperation [17,18]. The antibacterial capabilities of poly(vinyl alcohol) composite nanofibers combined with nanoscale Ge and SiO_2_ particles were discovered to be triggered by FIR radiation with wavelengths ranging from 5 to 20 µm [19]. Another organic product that can release FIR radiation is bamboo charcoal. It is also possible to produce far-infrared radiation from organic materials such as bamboo charcoal. These properties make activated charcoal an excellent absorbent and electromagnetic shielding and filter for a wide variety of applications. A previous study discovered that the addition of bamboo charcoal powder into chitosan films promoted water and oil absorption while decreasing mechanical characteristics at greater doses of bamboo charcoal [20]. Agricultural wastes such as oil palm shells, coconut shells, and nutshells, particularly macadamia nut shells, can also be used to make functional charcoal.

The macadamia nut-in-shell, which accounts for over 70% of macadamia nut processing waste, is extremely difficult to break down. In our earlier work, the macadamia nut-in-shell was transformed into a functional carbon by applying high-temperature carbonization at 1200 °C to generate a distinct material defined as Macca carbon (MC) [21]. MC has great porosity (about 350 m^2^/g), is rich in natural minerals, and has a large quantity of anions and the capacity to generate FIR radiation with wavelengths ranging from 6 to 20 µm and a deep-penetrating ability of around 4–5 cm. Based on its FIR-emitting capability, MC might be used as an antibacterial material in food packaging applications. As a reason, the purpose of this research would be to assess the efficiency and performance of the manufactured LDPE/MC composite film.

## 2. Materials and Experimental Method

### 2.1. Materials

Macca carbon (MC) powder was generously donated by Maccaya Co., Ltd. (Salaya, Nakhon Pathom, Thailand) as the FIR-generated material. The MC manufacturing method is described elsewhere [21]. The Plastic Institute of Thailand (Khlong Toei, Bangkok, Thailand) generously sponsored the virgin low-density polyethylene (LDPE) pellets (InnoPlus LD2426H Grade). Shaking was used to distribute the MC powder with the LDPE pellet at varied MC loadings of 0, 0.5, 1, 3, and 5% by weight. The LDPE/MC composites were manufactured by feeding a mixture of the LDPE and MC powder into a twin-screw extruder at a 5 Hz rotating speed, with temperature control in each zone as follows: hopper and feeding zones: 100–140 °C, compression zone: 150–170 °C, melting and die-head zones: 180 °C. They were then maintained at room temperature for 24 h before being dried in a hot oven at 70 °C for 1 h before being blown into a film with an average thickness of 55 ± 5 µm to accommodate physical property evaluation. The LDPE/MC-0, LDPE/MC-0.5, LDPE/MC-1, LDPE/MC-3, and LDPE/MC-5 composite film samples were designated after the MC powder loadings of 0, 0.5, 1, 3, and 5% by weight, respectively.

### 2.2. Characterization

#### 2.2.1. Fundamental Characterization of MC Powder

The thermal gravimetric analyzer (TGA/SDTA 851e, METTLER, Columbus, OH, USA) was used to perform the proximate analysis, expressed in moisture, volatile matter, and fixed carbon. Approximately 10 mg of MC powder was heated from room temperature to 110 °C in a nitrogen environment until dehydration was achieved, which was followed by decomposition at 850 °C for 10 min to quantify the quantity of volatile matter. The environment was then adjusted to become oxidizing. The specimen was cooled to 800 °C and kept for 3 h; then, it was placed in a desiccator until its weight remained unaltered. During this time, the weight loss was caused by the interaction of fixed carbon with oxygen. An adsorption assessment of MC, determined by nitrogen adsorption at −196 °C, led to evaluating their BET surface area, which was derived from the adsorption isotherms using the BET equation, following ASTM D4820. A scanning electron microscope (SEM S-2500, HITACHI, Tokyo, Japan) was employed to verify the morphology and particle size of MC. In this analysis, MC was coated with Pd-Pt.

#### 2.2.2. Morphological Characterizations

The dispersion of MC in the LDPE/MC composite film was investigated using optical microscopy (OM; Nikon, model Eclipses E200, Tokyo, Japan) and a Quanta 250 W7 scanning electron microscope (SEM; FEI Co., Brno, Czech Republic). For SEM analysis, all samples were cut and mounted onto an aluminum stub and coated with a thin layer of gold for 60 s in a vacuum chamber before imaging.

#### 2.2.3. Thermal Properties

Differential scanning calorimetry (DSC; Q10 TA instrument, New Castle, DE, USA) was used to determine the thermal characteristics of LDPE/MC composite films at a heating rate of 10 °C/min under a nitrogen atmosphere. The thermal stability of an LDPE/MC composite was investigated using thermal gravimetric analysis (TGA; TG 209 F1 Libra^®^, NETZSCH, Selb, Germany). Samples weighing 10 mg were scanned at temperatures ranging from 30 to 700 °C under a nitrogen atmosphere at a rate of 20 °C/min. Values of onset degradation temperature and peak temperature were analyzed from TGA curves.

#### 2.2.4. Mechanical Properties

The mechanical properties of LDPE/MC composite films were determined according to ASTM D882 utilizing a universal testing machine (Instron, model 5566, Norwood, MA, USA). All measurements were made on ten samples at a temperature of 25 °C and a relative humidity of 75%. The extension rate was set to 100 mm/min, and the load cell employed was a 20 kgf load cell with an 8 mm gauge length. 1.5 cm × 15 cm was the dimension of the sheet utilized.

#### 2.2.5. FIR Irradiation Characteristics

The FIR emission power generated by the MC in LDPE/MC composite films was evaluated in the wavelength range of 6–20 µm at 25 °C, using a Thorlabs S310C thermal power sensor machine connected with a PM100D energy meter as a detector, with data averaging 20,000 at a time interval of 4.0 s.

#### 2.2.6. Barrier Properties

The LDPE/MC composite film thickness of 55 ± 5 µm was used. The oxygen transmission rate (OTR) and the water vapor transmission rate (WVTR) of the composite films were analyzed using an OTR 8001 oxygen permeability tester and a WVTR 7001 water vapor permeation analyzer (both from Systech Instruments, Johnsburg, IL, USA). A barrier LDPE/MC composite film was sealed between a chamber holding oxygen and a chamber null and void of oxygen. According to ASTM D3985 Test Procedure, a coulometric device monitors the oxygen transported through the material at 23 °C and 0% relative humidity. Under phosphorous pentoxide, WVTR experiments were carried out with an LDPE/MC composite film sealed between a wet chamber and a dry chamber (P_2_O_5_). According to ASTM F1249, a pressure modulated sensor monitors moisture transferred through the material when it is conducted at 90% relative humidity and 37.5 °C. Both tests have been performed in triplicate, and the results were averaged.

#### 2.2.7. Light Transmission and Opacity Properties

The light transmittance and opacity of the LDPE/MC composite films were measured in triplicate using a Hunter Laboratories (ColorQuest^®^XE) CIE colorimeter equipped with an Illuminant D65 light source. The LDPE/MC composite film thickness of 55 ± 5 µm were cut into rectangles and inserted into the instrumental interior. The instrument was conditioned in accordance with ASTM D-1003 at a temperature of 23 °C and relative humidity of 50%.

#### 2.2.8. Antimicrobial Activities

Antibacterial behaviors of LDPE/MC composite films were assessed following JIS Z 2801 [22]. The microorganisms studied were strains of *Escherichia coli* (*E. coli*), a Gram-negative target organism, and *Staphylococcus aureus* (*S. aureus*), a Gram-positive target organism. At 38 °C for 24 h, *E. coli*, and *S. aureus* were grown separately on a MacConkey agar plate and a trypticase soy agar plate. A single colony was adjusted to a 10 mL volumetric flask using either nutritious broth or trypticase soy broth. Both sides of the LDPE/MC composite films were sterilized using a UV-A light for 2 h and were injected with the cultured broth. Next, the microbial inoculum was coated with a thin sterile film, and the samples were incubated in a chamber at 38 ± 1 °C and 90% humidity for 24 h. After incubation, the samples were rinsed with 30 mL of neutralizer, and the colony-forming units (CFUs) were calculated. The percentage of microorganism reduction (% R) was determined using the following equation:% R = (B − C)/B × 100
where B represents the number of viable microbial cells in the control LDPE/MC-0 sample and C corresponds to the number of viable microbial cells in the LDPE/MC composite film after 24 h. The average of the outcomes of three separate tests was determined.

#### 2.2.9. Storage Tests

Plastic bags fabricated from LDPE/MC composite films were used to explore the freshness of lettuce and strawberries after seven days of storage at room temperature (28 °C) and 65% relative humidity. The lettuce and strawberry specimens were meticulously chosen for their consistent size and color and absence of mechanical damage, and aesthetic abnormalities. The rot progression behavior of the specimen was visually monitored during the test period.

## 3. Results and Discussion

### 3.1. Fundamental Characteristics of MC Powder

The fundamental data of MC powder acquired from thermogravimetric analysis employed in this study, including moisture, volatile matter, and fixed carbon are 1.74, 2.58, and 87.97% by weight, respectively. These data supported that the purity of MC powder is quite high enough for further use. The adsorption characterization yielded an average BET surface area of 748.6 m^2^/g for MC. The surface area is high enough for absorption purposes. Figure 1 depicts SEM micrographs of MC powder taken at 1500 and 4000 expansion times. The average particle size of MC powder was found to be 5.7 µm with polygonal structures.

### 3.2. LDPE/MC Composite Film Fabrication

Melt-compounding twin-screw extrusion was used to prepare all of the LDPE/MC composite films, which were subsequently manufactured utilizing a blown film extrusion. Each specimen is a controlled specimen with a thickness of 55 ± 5 µm. To commence, a visibly magnified optical microscopic approach was used to evaluate the dispersion of MC powder in the LDPE/MC composite films, as seen in Figure 2. The black dots are MC powder spread throughout the LDPE/MC composite sheets. Although the MC powder was distributed adequately in all samples, adding more MC powder to the LDPE/MC composite films contributed to a denser dispersion. This finding shows that using excessively MC powder may not be required.

Figure 3 shows SEM micrographs of the LDPE/MC composite specimens. The surface of the pure LDPE specimen looked rather smooth. The surface of composites became rougher as the amount of MC powder added to the LDPE increased. The aggregation of MC powder and blooming to the surface of the composites were most visible at 3 and 5% MC. The poor interfacial adhesion between MC and LDPE yielded these results, which are consistent with the OM study. This evidence may cause a decline in mechanical characteristics as a consequence of rising MC powder content, which will be explored further below.

### 3.3. Thermal Properties

DSC analysis was used to study the effect of MC powder on the thermal properties of the LDPE/MC composite film. Table 1 shows the melting temperatures (T_m_), melting enthalpies (ΔH_m_), and crystallinity percentages (X_c_) of the LDPE/MC composite films. When the MC powder loading was increased, ΔH_m_ ascended significantly, while T_m_ values remained relatively steady. The percentage ratio of ΔH_m_ of sample and pure LDPE sample (293 J/g) can be used to calculate the X_c_ value. It is obvious that the crystallinity percentages of the LDPE/MC composite films increased from 37.8% to 51.3% when the MC content increased by up to 5%. This evidence suggests that the MC powder induces crystallinity directly, resulting in the increased film strength, as will be discussed later.

TGA analyses of the LDPE/MC composites were also carried out, as can be seen in Figure 4 and Table 1. It is noticeable that all of the composite films showed a similar one-step decomposition behavior at around 400–530 °C, demonstrating that the existence of MC powder had no considerable impact on the thermal characteristic of the LDPE/MC composite films. However, the losing weight temperature at 5% and 10% (T_5%_ and T_10%_) was slightly elevated as MC content increased, implying that the contribution of MC powder may enhance the thermal stability of the LDPE film. Due to the obvious strong incorporation of MC powder in the LDPE matrix, the percentage of residues at 700 °C by TGA analysis was also fairly raised, and MC 3% and 5% exhibited relatively high values. These results imply that a higher amount of MC powder would lead to the residue after degradation, so the optimum amount should be as low as possible while still affecting the other properties of the composite film. (Note: T_5%_ and T_10%_ of MC powder were 315.8 and 618.9 °C, respectively.)

### 3.4. Mechanical Properties

Figure 5 illustrates the mechanical characteristics of the LDPE/MC composite films as determined using a universal testing equipment. It is obvious that varying the amount of MC powder used as the FIR generator in the virgin LDPE has a direct effect on the tensile strength and break elongation. Tensile strength of the LDPE/MC composite films with MC powder concentrations of 0.5, 1, and 3% by weight increased approximately 4.6, 7.3, and 22%, respectively. In contrast, the LDPE/MC composite films with a higher concentration of MC powder at 5% by weight contributed to a 2% reduction in tensile properties compared to the virgin LDPE. Increases in the MC powder content to 0.5, 1, 3, and 5% by weight resulted in about 17, 33, 73, and 83% decreases in elongation at break, respectively, compared to virgin LDPE film. This finding means that increasing the amount of MC powder increases strength up to a point while lowering elongation at break. On the basis of these data, it may be assumed that a concentration of 0.5% by weight should suffice to maintain the distinguishing features of the LDPE/MC film.

### 3.5. FIR Emission

The FIR emission power spectra of the LDPE/MC composite films at various MC powder concentrations in the wavelength range of 6–20 µm at 25 °C is shown in Figure 6. The virgin LDPE film exhibited essentially negligible irradiation at the frequencies of interest. However, increasing the MC content in LDPE/MC composite films increased the FIR emission power. This work demonstrated that MC is capable of being embedded in LDPE as a composite material. Since FIR is a kind of thermal energy, increasing the quantity of MC in the LDPE/MC composite results in an increased heat conduction. This result indicates that the LDPE/MC composite with 0.5% by weight MC was sufficiently functioning to provide FIR.

### 3.6. Barrier Properties

The oxygen transmission rate (OTR) and water vapor transmission rate (WVTR) are typically used to describe the barrier characteristics of composite films. The principal goal of these specifications for materials used in food packaging is to keep the product’s flavor while shielding it from external contamination. The OTR and WVTR of the LDPE/MC composite films are depicted in Figure 7. In comparison to the control LDPE, the LDPE/MC composites had a greater OTR and WVTR, demonstrating that they had a reduced barrier property. In other words, the oxygen and vapor permeations are greater in the experimental condition than in the control condition. This finding suggests that the existence of MC may enhance the barrier characteristics well.

### 3.7. Light Transmission and Opacity Properties

Figure 8 illustrates the light transmission and opacity of the LDPE/MC composite films. Light transmission refers to the percentage of incident light that travels through a film. Opacity is the property of a thin and clear material that allows it to conceal the surface behind it; it is the inverse of transparency. Opacity is defined as the ratio of the material’s reflectance when backed by a black substrate to the relectance of material when backed by a white substrate. Thus, increasing the MC dose increased opacity and decreased light transmission and *vice versa*. According to the findings, the LDPE/MC composite with 0.5% by weight MC appears to satisfy the requirements for adequate packing characteristics at a level comparable to those of the virgin LDPE film.

### 3.8. Antimicrobial Activities

Table 2 shows the results of testing the antibacterial activity of LDPE/MC composite films using the JIS Z 2801 procedure. *E. coli* and *S. aureus* were chosen as the target Gram-negative and Gram-positive bacteria, respectively, in this investigation. As a control, LDPE/MC-0 or the virgin LDPE film was utilized to compare to other specimens treated with various MC powder dosages. According to Table 2, the reduction percentage of microorganisms (% R) was highly dependent on the MC powder dosage and microbial type. The greatest % R value was achieved as LDPE/MC-0.5 was used. In other words, the LDPE/MC composite films containing 0.5% by weight MC demonstrated a substantial antibacterial activity against *E. coli* and *S. aureus*. In our findings, the antibacterial properties of the LDPE/MC composites deteriorated when the MC loading increased by over 0.5%. As seen in Figure 6, an increased MC loading causes an increased FIR emission, which is known to raise the target materials’ temperature or living beings [17,18,21]. If the temperature is raised to the appropriate level, bacteria will proliferate faster than they die. As a result, only the appropriate concentration of MC powder, approximately 0.5%, can directly serve as a bactericide in conjunction with FIR generated by MC powder [23,24,25].

As described in the previous section, MC powder may effectively generate the FIR in the wavelength range of 6–20 µm, promoting antibacterial and deodorization and perhaps increasing shelf-life [23]. However, the actual mechanism through which FIR materials exhibit such high antibacterial activity and their usefulness in preserving food quality have not been well investigated or understood.

### 3.9. Storage Tests

Plastic bags fabricated from LDPE/MC composite films were used to determine the shelf-life of food packaging in terms of freshness. In this study, lettuces and strawberries were chosen, and the decaying time was observed for one week. A plastic bag produced from LDPE/MC films with varying concentrations of MC powder was used to retain lettuce and strawberries at room temperature (28 °C, not in the refrigerator) for a week. Figure 9 and Figure 10 show the representations of each experiment after they were stored for 3, 5, and 7 days, respectively. Each experiment included a visual examination of triplicate samples. It is evident that lettuces packed with LDPE/MC-0, LDPE/MC-1, LDPE/MC-3, and LDPE/MC-5 began to rot on the third day. The same results were observed in the case of strawberries.

Interestingly, only LDPE/MC-0.5 preserved the freshness of lettuces longer for 7 days, despite the fact that there was no difference in visual appearance from day 1 to day 7, and only LDPE/MC-0.5 retained the freshness of strawberries for 5 days at 28 °C, not in the refrigerator. This storage test, antimicrobial test, FIR test, and mechanical property tests revealed that a concentration of 0.5% by weight MC powder is perfectly adequate for food quality preservation and effectively extends food shelf-life without refrigeration. This point is because the FIR released by the MC powder may impact cell metabolism and enhance cell rejuvenation [17].

## 4. Conclusions

Various concentrations of MC powder were mixed with LDPE in this investigation, i.e., 0, 0.5, 1, 3, and 5% by weight. The fundamental characteristics of the LDPE/MC composite film were determined. It was discovered that increasing the concentration of MC powder resulted in the improved film strength but decreased elongation qualities due to the proportionate incidence of high crystallinity. In terms of FIR emission power, all MC powder concentrations in the LDPE/MC composite films outperformed the virgin LDPE film. However, the most appropriate doses for constructing the LDPE/MC composites were 0.5% weight based on the barrier and light transmission characteristics. Finally, storage and antibacterial activity tests indicated that a 0.5% by weight MC powder in the LDPE/MC composite films resulted in the maximum percentage of microorganism decrease. As a consequence, it is feasible to conclude that MC powder at a concentration of 0.5% by weight is the best concentration to combine with LDPE to create the unique functional packaging film that efficiently protects food quality and extends its shelf-life. This study shows that LDPE/MC composite films are innovative and promising materials for implementation in food manufacturing.

## Figures and Tables

**Figure 1 polymers-14-01794-f001:**
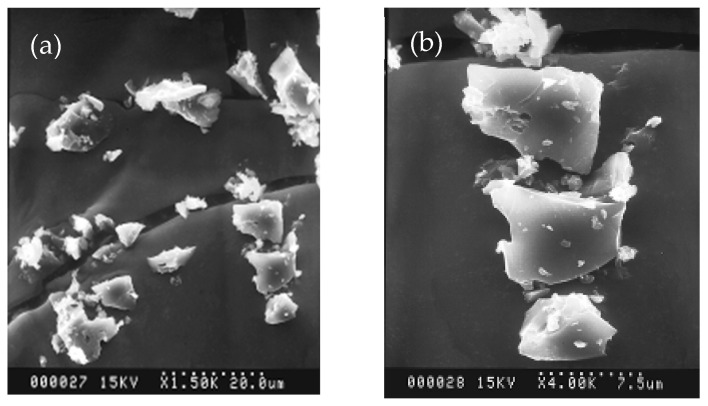
SEM micrographs of MC powder (**a**) ×1500 and (**b**) ×4000.

**Figure 2 polymers-14-01794-f002:**
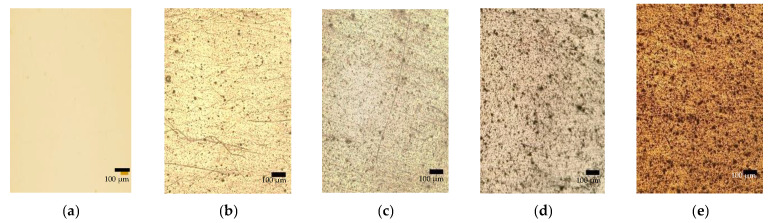
OM images of (**a**) LDPE/MC-0, (**b**) LDPE/MC-0.5, (**c**) LDPE/MC-1, (**d**) LDPE/MC-3, and (**e**) LDPE/MC-5.

**Figure 3 polymers-14-01794-f003:**
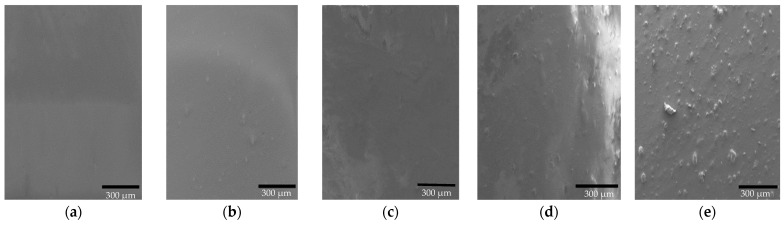
SEM micrographs of (**a**) LDPE/MC-0, (**b**) LDPE/MC-0.5, (**c**) LDPE/MC-1, (**d**) LDPE/MC-3, and (**e**) LDPE/MC-5.

**Figure 4 polymers-14-01794-f004:**
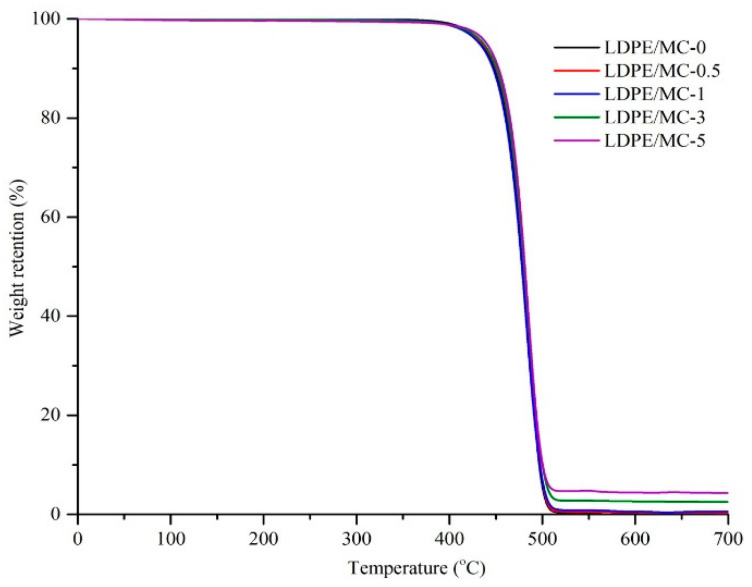
TGA thermograms of LDPE/MC composites films at various MC concentrations.

**Figure 5 polymers-14-01794-f005:**
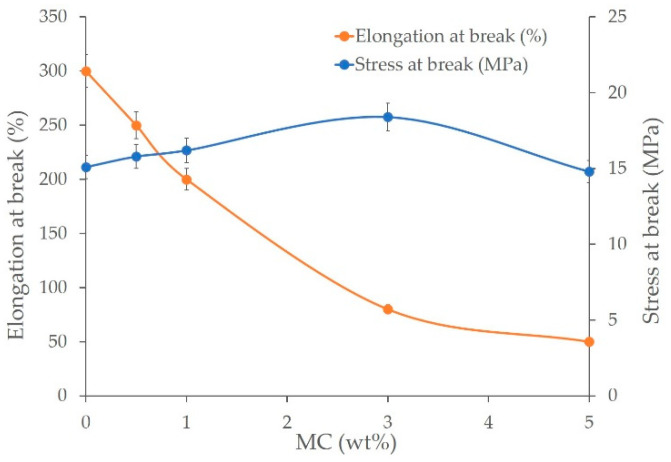
Tensile strength and elongation at break of the LDPE/MC composite films at various MC powder concentrations.

**Figure 6 polymers-14-01794-f006:**
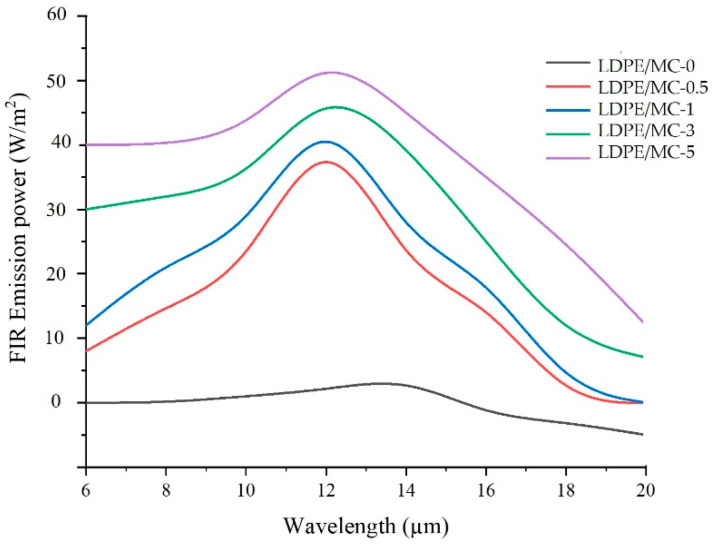
FIR emission spectra of the LDPE/MC composite films at various MC powder concentrations.

**Figure 7 polymers-14-01794-f007:**
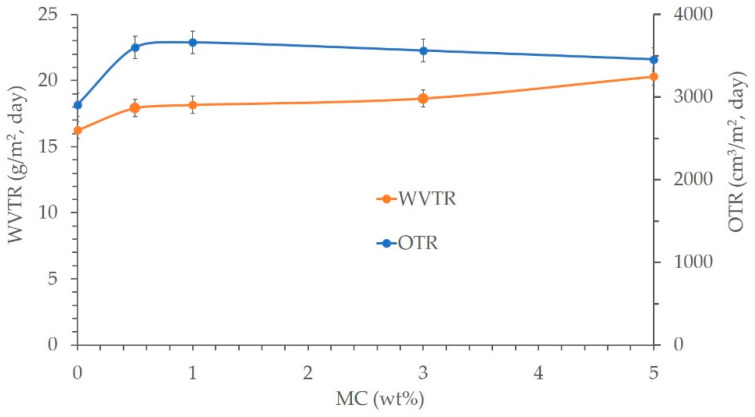
The oxygen transmission rate (OTR) and water vapor transmission rate (WVTR) as the barrier properties of various LDPE/MC composite films.

**Figure 8 polymers-14-01794-f008:**
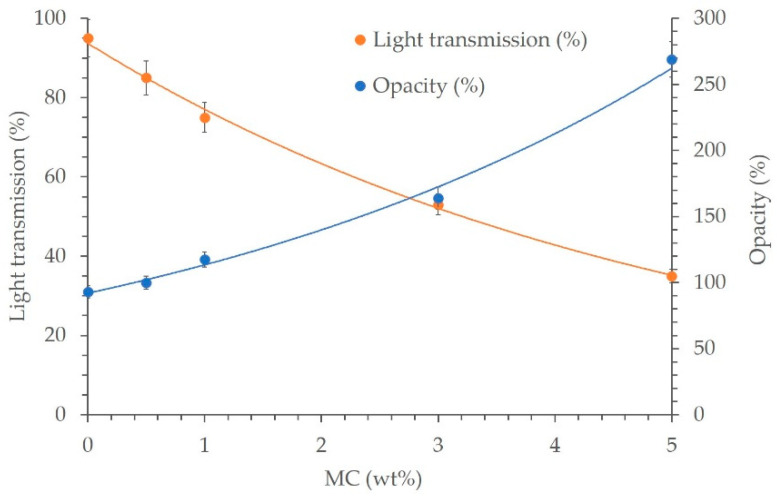
Light transmission and opacity of various LDPE/MC composite films.

**Figure 9 polymers-14-01794-f009:**
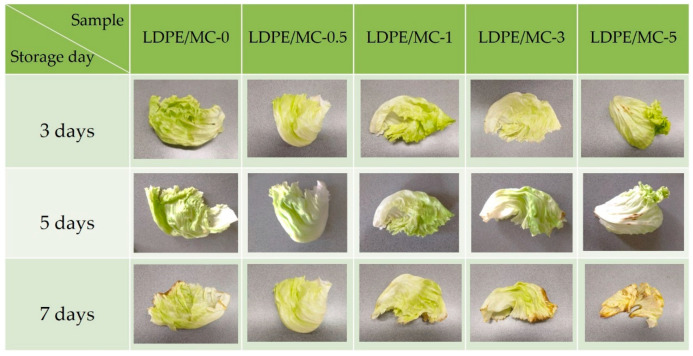
Photographs of lettuces packed with various LDPE/MC composite films for 3/5/7 days at 28 °C.

**Figure 10 polymers-14-01794-f010:**
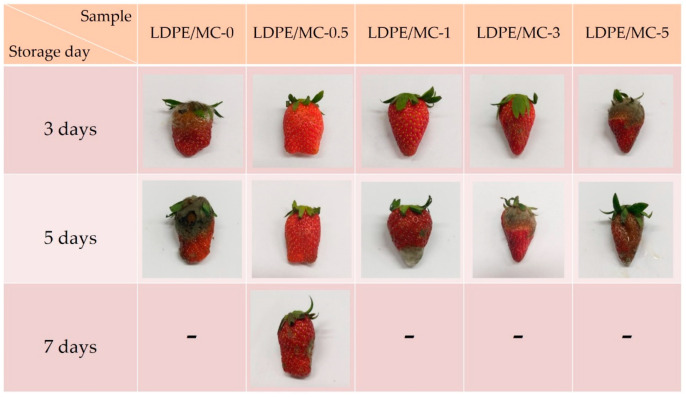
Photographs of strawberries packed with various LDPE/MC composite films for 3/5/7 days at 28 °C.

**Table 1 polymers-14-01794-t001:** Thermal properties of LDPE/MC composite films by TGA and DSC analyses.

Method	DSC			TGA		
Sample	T_m_ (°C)	ΔH_m_ (J/g)	X_c_ (%)	T_5%_ (°C)	T_10%_ (°C)	Residues (%)
LDPE/MC-0	109.03	110.9	37.8	435.4	449.6	0.05
LDPE/MC-0.5	109.25	128.7	41.9	436.2	450.0	0.32
LDPE/MC-1	109.51	135.5	43.2	432.4	446.9	0.55
LDPE/MC-3	109.79	145.6	49.7	436.9	451.0	2.50
LDPE/MC-5	109.90	150.5	51.3	440.4	453.4	4.33

**Table 2 polymers-14-01794-t002:** Antimicrobial Activities of LDPE/MC Composite Films.

	Type	*E. coli*	% R	*S. aureus*	% R
Sample					
LDPE/MC-0		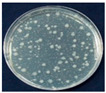	-	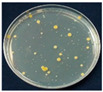	-
LDPE/MC-0.5		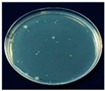	90.8	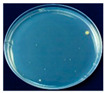	95.2
LDPE/MC-1		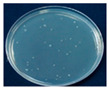	81.4	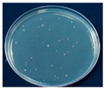	72
LDPE/MC-3		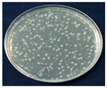	7.3	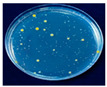	52.6
LDPE/MC-5		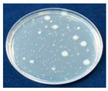	2.6	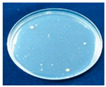	30.1

## Data Availability

Not applicable.

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
