# Peer review of "A Low-Density Polyethylene (LDPE)/Macca Carbon Advanced Composite Film with Functional Properties for Packaging Materials"

_polymers, 2022, doi:10.3390/polym14091794_

Round 1

Reviewer 1 Report

The manuscript is current, it devotes the currently observed trends in the preparation of polymeric composite materials with natural fillers - in this case, LDPE-based packaging material modified with Macca carbon (MC) powder by different concentrations.

In the introduction the authors introduces frequently used inorganic fillers for the food packaging based and their reaction mechanisms which leds to prolong the freshness of food mainly based on FIR radiation and FIR emission qualities.

The authors sufficiently described the basic conditions of the characterisation methods. I have the following comments on the work:

The basic characteristics of Macca carbon (MC) powder must be added to the manuscript. It is stated that the procedure for their preparation has already been published in another paper. Here I lack any powder characteristics, eg size of powders, specific surface area, purity (evidenced by X-ray or XRFS analysis).

The structural characteristics of LDPE and LDPE / MC composites are insufficient. Figure 1 does not show the scale (magnification) and at the same time the text lacks any description of structural changes. What are the black dots on the OM images of Macca carbon (MC) powder or pores / cavities? How was the porosity of the LDPE / MC composites?

What is the expected mechanism of action of Macca carbon (MC) powder on the resulting mechanical properties, especially Young's modulus? Substantiate your claim "… that a concentration of 0.5 percent by weight should suffice to maintain the distinguishing features of the LDPE / MC composites." After all, there is insufficient structural characteristics of LDPE / MC composites.

It is not clear enough from the text, when were the MIC values evaluated. Is it after 24 hours? Given the expected long-term use of packaging material (5 and 7 days), it would be appropriate to supplement with antimicrobial tests on these days, ie. monitor the long-term antimicrobial activity of the LDPE / MC composite.

Reviewer 2 Report

In this work, LDPE/Macca carbon composites were prepared, and their properties were studied. In spite of some valuable results, there are some problems that should be clarified and more experiments and discussion should be supplied before further consideration.

  1. Please supply the size and morphology of MC.
  2. In page 3, line 94-98, the characterization of optical microscopy is not belong to thermal properties.
  3. The authors used optical microscope to evaluate the dispersion of MC in LDPE. It is feasible. However, analyzing the compatibility between MC and LDPE is also of great importance. It is better to use SEM to study the interfacial adhesion between MC and LDPE, which is a key factor that may influence the mechcanical and barrier performance.
  4. In practical applications, the thermal stability of materials is also very important. Please supplement the thermal decomposition behaviors of MC, LDPE and LDPE/MC composites.
  5. In section 3.7, LDPE/MC-0.5 showed good antimicrobial performance. However, with the increase of MC loading, the composites’ antimicrobial performance deteriorated. Why? Please clarify.

Round 2

Reviewer 1 Report

The authors of the manuscript made the required changes and additions to the necessary text. I agree with publishing.

Reviewer 2 Report

This work has been improved by the authors. It can be accepted for publication.